# Spatio-Temporal Dynamics in Grasslands Using the Landsat Archive

Astrid Vannoppen [1,*], Jeroen Degerickx [1], Niels Souverijns [1] and Anne Gobin [1,2]

1   Vlaamse Instelling voor Technologisch Onderzoek NV, 2400 Mol, Belgium; jeroen.degerickx@vito.be (J.D.); niels.souverijns@vito.be (N.S.); anne.gobin@vito.be (A.G.)
2   Department of Earth and Environmental Sciences, Faculty of BioScience Engineering, University of Leuven, 3001 Leuven, Belgium
*   Correspondence: astrid.vannoppen@vito.be

**Abstract:** Grasslands are an important biotope in Europe, not only because they are widespread, but also because they provide valuable ecosystem services. The ecological value of a grassland parcel is directly proportional to the number of uninterrupted years of grassland cover. However, the area of long-term grassland (i.e., grassland of 5 years or older) is decreasing, limiting its ability to provide ecosystem services. To prevent the further disappearance of long-term grasslands, Europe developed an agricultural policy instrument in 2003 to protect grasslands of 5 years or older. Nature policy instruments aim to protect grasslands that have existed for more than 10 years to support their high environmental value. However, there is currently no multi-annual information on the location and age of grasslands at a high spatial and temporal resolution, which makes it difficult to assess the effectiveness of the current grassland protection regulations. Multi-annual satellite-based land cover classification can provide a solution for grassland area and age monitoring, which we tested by producing a series of Landsat-based land cover classification maps from 2005 to 2019 for the region of Flanders, Belgium. Historical land cover classification maps proved useful for evaluating past and present planning and policy to ensure grassland conservation, linking spatial and temporal changes in the area of long-term grasslands with policy changes and landscape dynamics. We were able to locate grasslands that were grassland between 2005 and 2014 but were converted to arable land between 2015 and 2019, identify the year in which these grasslands were converted to arable land, and demonstrate regional differences in the conservation of long-term grassland aged 5–9 years and 10 years or more. Long-term grassland aged 10 years or more disappeared faster in urban than in rural areas in Flanders between 2014 and 2019. Our study shows that multi-annual high-resolution satellite imagery provides objective and quantitative information on long-term grassland to support climate, agricultural, environmental, and nature policies.

**Keywords:** grassland; Landsat; land cover; permanent grasslands; historical grasslands; land cover classification; LULCC; cover dynamics; policy; random forest; hidden Markov model

## 1. Introduction

Grasslands are an important biotope in Europe, covering 18.7% of Europe in 2018 according to the Land Use-Land Cover Area frame Survey (LUCAS) survey [1]. In addition to their widespread presence, grasslands provide a wide range of ecosystem services that are essential for human well-being [2–6]. In agricultural areas, these biotopes represent a major source of fodder for livestock, as such being an integral part of our meat and dairy production systems [3,4]. Aside from this, grasslands, and especially long-term grasslands (i.e., grassland of 5 years or older) can store substantial amounts of carbon, mainly in soil organic matter [5]. Globally, the carbon sequestration potential of grassland ecosystems older than 10 years is estimated to be 0.01–0.3 Gt C/year, which would offset up to 4% of global greenhouse gas (GHG) emissions [7]. Grasslands are therefore often proposed

as part of a climate change mitigation strategy. In addition, their permanent nature, as opposed to temporary croplands, reduces water runoff, and thus soil erosion [2]. This reduction in soil erosion is linked to other regulating services, such as water regulation, carbon storage and soil fertility [2,3]. Finally, grasslands are important for biodiversity conservation, as they support about one third of the total higher plant flora in Central Europe [8,9]. They are essential for a wide range of farmland birds, butterflies, beetles, and other animals [8].

Unfortunately, the degradation and conversion of grasslands is putting pressure on the provision of these important ecosystem services [2,3,6,10]. In the EU-6, approximately 7.1 million ha (or 30% of the 1967 level) of long-term grasslands disappeared between 1967 and 2007 [11]. The driving forces of grassland loss are agricultural intensification, land abandonment leading to succession and forestation, and the expansion of human settlements into agricultural land [10,11]. To prevent further loss of grassland in agricultural areas, the European Union implemented the cross-compliance regulation in 2003, which requires farmers receiving support under the Common Agricultural Policy (CAP) to maintain grasslands of at least 5 years old, referred to in the CAP as permanent grassland [6,11].

Multi-annual data on the location of grasslands at a high spatial and temporal resolution are currently lacking in Europe. This makes it difficult to monitor the area, age, and quality of long-term grasslands. The LPIS geodatabases available in Europe provide annual data for agricultural grasslands. For non-agricultural grasslands, the triennial LUCAS surveys, which have been available since 2001, can be used for monitoring. However, as the LUCAS surveys only consist of in situ observations, which are not available on an annual basis, it is difficult to determine the age of disappearing grasslands. Satellite data can help address this information gap by providing long-term, consistent, and spatially continuous data on the characteristics of the Earth's surface. The advent of publicly available, long-term satellite archives, such as Landsat, and freely available cloud computing platforms has made it possible to monitor and assess changes in the spatial and temporal distribution of land cover from a country to a global scale using satellite-derived historical land cover classification maps [12–16]. For example, using the Landsat archive, it was possible to evaluate changes in grassland area from year to year, giving insight in livestock driven grassland and land cover dynamics [13,17]. Additionally, characteristics such as forage quality [18] and frequency and timing of mowing [19] can be derived from satellite data.

In northern Belgium, grasslands are prominently present in the rural landscape. According to the 2018 LUCAS survey [1], grasslands covered 26% of the land area in Flanders, Belgium, which is higher compared to the share in Europe (18.7%) [1]. Based on the land parcel identification system geodatabase (LPIS geodatabase), 35% (or 242,601.2 ha) of the agricultural land in Flanders was grassland in 2018 [20]. However, the agricultural landscape in Flanders is dynamic. Between 2008 and 2018, the agricultural grassland area decreased by 4.6%, or 11,630.2 ha in just 10 years [20]. The age of the agricultural grasslands that have disappeared is unclear, while their ecological value and ecosystem services are directly influenced by the number of consecutive years the field has been maintained as grassland [2,6]. In addition, the location and age of natural grasslands or grasslands in private/public gardens is largely unknown.

Considering the high proportion of grasslands in the landscape and the important ecosystem services they provide, the conservation of grasslands is highly desirable. However, information on the temporal and spatial changes of grasslands in Flanders is currently missing. This hampers the evaluation and adjustment of policies aimed at the protection of long-term grasslands and the continued provision of ecosystem services. The objectives of this manuscript are to (1) identify the location of grasslands in Flanders, both agricultural and non-agricultural, by producing land cover classification maps from 2005 to 2019, and (2) to demonstrate how these land cover maps can be used to (i) evaluate conversion of grassland to arable land, and (ii) evaluate quantitative changes in the area of long-term grasslands aged 5–9 years and 10 years or more.

## 2. Materials and Methods

### 2.1. Study Area

Our study area Flanders in Belgium is a densely populated, highly fragmented, and heterogeneous region (Figure 1). Around half of its surface area (51%) is covered by agriculture, 16% by infrastructure and 13% by forest [21]. Grasslands dominate the agricultural landscape in Flanders and occupied 35% (or 242,601.2 ha) of the agricultural area in 2018 [20].

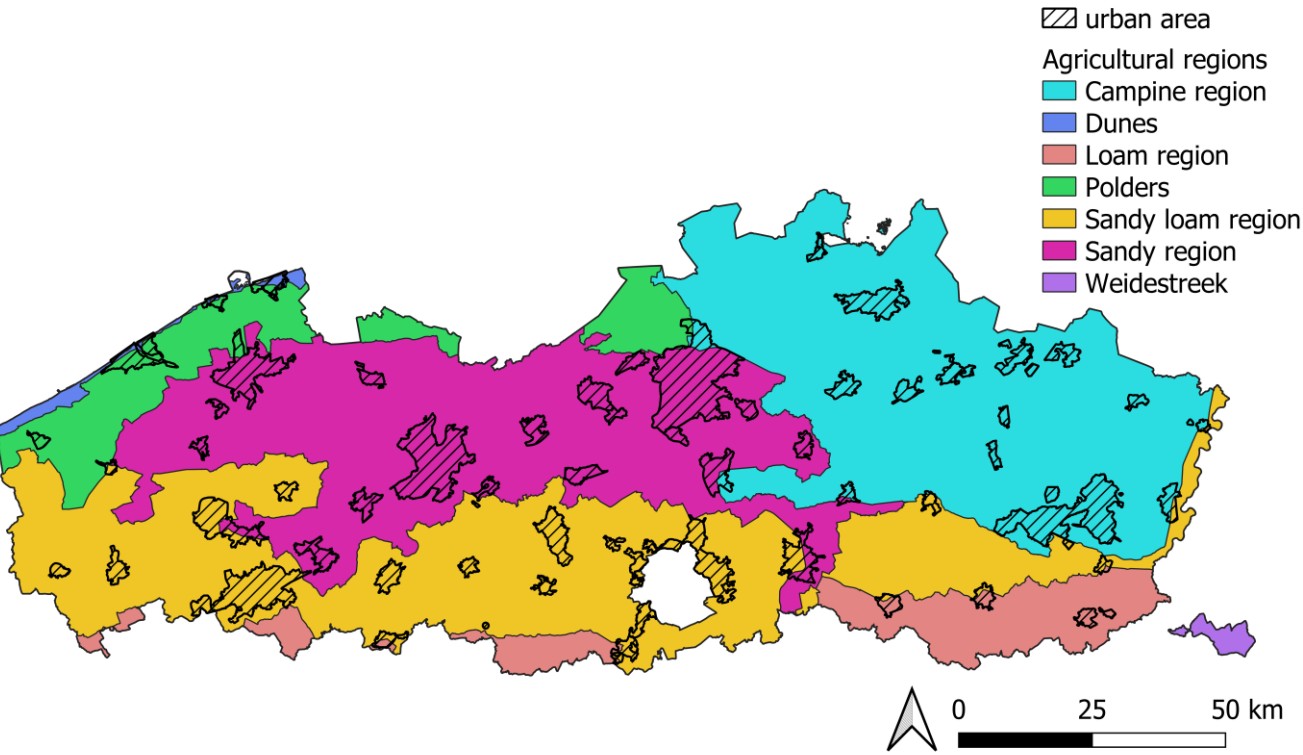

**Figure 1.** Seven agricultural regions of Flanders. Shaded polygons indicate urban areas and non-shaded polygons indicate rural areas in Flanders as defined in the spatial structure plan of Flanders of 2019.

### 2.2. Methodology for the Land Cover Classification Maps

Grassland dynamics in Flanders between 2005 and 2019 were investigated using historical land cover classification maps at 30 m resolution. The historical land cover maps were produced using a random forest classification algorithm processing the Landsat 5, 7 and 8 archive following the methodology of [12,16]. The entire Landsat archive from 2005–2019 (Landsat 5, 7 and 8) with images over Flanders was used. The locations of the scenes are listed in Appendix A. Collection 1 Tier 1 surface reflectance data of Landsat 5, 7, and 8 available on Google Earth Engine were used. Pixels affected by clouds were flagged by the internal C Function of the Mask (CFMASK) algorithm and removed. Multiple metrics were derived from the spectral information and directly used as input for the random forest classification algorithm. The metrics included descriptive metrics (mean, median, 10th and 90th percentile, and the 10th–90th percentile range), harmonic metrics, textural metrics, and topographic metrics, which together accurately describe the seasonal cycle of different land cover types. The seasonality of vegetation growth was captured by the harmonic metrics [16]. All calculations were performed in the Google Earth Engine [22].

The land parcel identification system (LPIS) geodatabase of Flanders [23] and the ground-validation land cover dataset of the International Institute for Applied Systems Analysis (IIASA) [24] of 2012 and 2015 were used as test and training datasets for the classification algorithm. The number of ground-truth classified pixels was 20,046 in 2012 and 20,135 in

2015. The datasets were split into 67% for training and 33% for testing. For each year from 2005 to 2019, land cover was classified into the following classes: cropland, grassland, forest, urban/bare and water. The 2012 and 2015 maps were validated using the test dataset (i.e., hold-out validation). The overall accuracies and the confusion matrix were calculated for the 2012 and 2015 maps. The 1 m resolution land cover map of Flanders of 2015 [25] was used in a post-classification step to further clean the generated maps by excluding pixels with >15% trees or buildings according to this high-resolution land cover map.

Classification maps for 2005–2012 were based on Landsat 5 and 7 imagery, while from 2013 the Landsat 7 and 8 archives were merged, resulting in a higher temporal resolution for the latter period. In addition, the Landsat 7 scan line corrector failed in 2003, resulting in data gaps and lower quality Landsat 7 data compared to Landsat 8 data [26]. The best available pixel approach from [27] was applied to account for the lack of good quality data from 2005–2012. Due to these inherent differences in input data quality between years, the post-classification hidden Markov model was applied to improve the consistency between maps of different years and remove spurious land cover class transitions between 2005 and 2014, thereby ensuring historical consistency in the classification [12].

*2.3. Evaluation of Grassland Area Based on Land Cover Classification Maps*

The land cover classification maps produced from 2005 to 2019 were used to evaluate the conversion of grassland to arable land. In addition, the maps allowed us to evaluate trends in the area covered by long-term grasslands. Long-term grassland aged 5–9 years and 10 years or more were evaluated separately. This was conducted by identifying pixels that were classified as grassland for five to nine consecutive years and ten or more consecutive years between 2005 and 2019. Changes in grassland between 2005 and 2019 were examined in urban versus rural areas and in the seven agricultural regions of Flanders (Figure 1).

**3. Results**

The overall accuracies for the 2012 and 2015 land cover maps were equal to 85.3% and 88.8%, respectively. The confusion matrices for both maps are presented in Table 1. For grassland, an accuracy of 72.7% was achieved in 2012 and 83.8% in 2015 (Table 1). For the classes bare/urban and water, the accuracy was not good when they were considered as separate classes, probably due to the lack of training data for the water class. However, when the classes urban/bare and water were merged good accuracies were achieved. Since the aim of the historical land cover classification maps was to identify grassland, the low accuracy for the classes bare/urban and water was not considered a limiting factor for further analysis.

A snapshot of the historical land cover classification maps for the years 2005, 2010, 2015 and 2019 is presented in Figure 2. This snapshot shows how the historical land cover classification maps can be used to identify spatial shifts in the grassland class between 2005 and 2019. Forests are not visible in the classification maps because pixels with >15% trees or buildings according to the 2015 1 m resolution land cover map were excluded in a post-classification step. From a visual inspection of the historical land cover classification maps, it was clear that the temporal (year-to-year) variability of the classification results was higher from 2015 to 2019 than from 2005 to 2014. This could be related to the hidden Markov model applied from 2005 to 2014, which eliminated some of the year-to-year variability in the land cover classification. This should be considered when looking at the annual land cover changes in the historical land cover maps.

By visually comparing the produced land cover maps with the available high resolution orthophotos for 2012–2021 [28], we could conclude that we were not only able to identify agricultural grasslands, but also correctly classify natural grasslands, football pitches, large gardens, grass strips in airports, etc. (Figure 3). Thus, the developed historical land cover classification maps provide additional information on grasslands compared to the LPIS geodatabase, which only includes agricultural grasslands.

**Table 1.** Confusion matrix for the historical land cover classification maps of 2012 and 2015.

| **2012 Historical Land Cover Classification Map** | | | | |
|---|---|---|---|---|
| Classification result | Ground truth | | | |
| | Forest | Grassland | Arable land | Urban/bare/water |
| Forest | 1663 | 39 | 39 | 0 |
| Grassland | 2 | 4577 | 1006 | 43 |
| Arable land | 37 | 1680 | 9893 | 83 |
| Urban/bare/Water | 0 | 4 | 12 | 968 |
| **2015 historical land cover classification map** | | | | |
| Classification result | Ground truth | | | |
| | Forest | Forest | Forest | Forest |
| Forest | 1715 | 38 | 19 | 99 |
| Grassland | 2 | 5325 | 967 | 35 |
| Arable land | 9 | 989 | 9894 | 82 |
| Urban/bare/Water | 0 | 3 | 15 | 943 |

The historical classification maps allowed us to assess changes in grassland by analyzing the sequence of land cover from 2005 to 2019. According to our results, 15% of Flanders (2104 km$^2$) was classified as grassland every year from 2005 to 2019. In 3% of Flanders (449 km$^2$), grassland that had been grassland since 2005 was converted to arable land between 2015 and 2019, with 27%, 22%, 20%, 16%, and 15% of this grassland conversion occurring in 2017, 2016, 2019, 2015, and 2018, respectively. The location of grassland conversion areas can be analyzed using the historical land cover classification maps (Figure 4).

Pixels classified as grassland for 5 up to 9 consecutive years from 2005 to 2019 were identified using the historical land cover classification maps. This resulted in maps from 2014 to 2019 showing the location of long-term grassland aged 5–9 years old. The evolution of long-term grassland aged 5–9 years old was calculated using these maps (Figure 5). The area of long-term grassland aged 5–9 years old derived from the historical land cover maps in Flanders is minimal. Therefore, we will focus on long-term grassland aged 10 years or more, hereafter referred to as long-term grassland.

Pixels classified as grasslands for at least 10 consecutive years from 2005 to 2019 (hereafter referred to as long-term grasslands) were identified using the historical land cover classification maps. This resulted in maps from 2014 to 2019 showing the location of long-term grasslands (Figure 6) and allowed changes in long-term grassland between 2014 and 2019 to be detected (Figure 7). Between 2017 and 2019, 18% of Flanders was covered by long-term grassland. The area of long-term grassland decreased significantly from 2014 (2866 km$^2$ or 21% of Flanders) to 2019 (2105 km$^2$ or 15% of Flanders). The decrease in long-term grassland was less in the period 2017 to 2019 than in the period 2014 to 2017.

On average, there was more long-term grassland in rural areas (i.e., 19%) than in urban areas (i.e., 9%) between 2014 and 2019 (Figure 8). The long-term grassland area decreased in both urban and rural areas of Flanders between 2014 and 2019, but the decrease was slightly higher in urban areas (16%) than in rural areas (15%) of Flanders.

The area of long-term grasslands varied in the seven agricultural regions of Flanders (Figure 9). The average area of long-term grasslands between 2014 and 2019 was highest in the Weidestreek region (34%) and lowest in the Loam region (18%). The long-term grassland area decreased in all the seven agricultural regions of Flanders between 2014 and 2019, being lowest in the Weidestreek (7%) and highest in the Loam region (16%). The typical grassland landscape of the Weidestreek is therefore not disappearing at an alarming rate. However, there was still a decrease in long-term grassland in this region,

which should be monitored to prevent the disappearance of the typical Weidestreek grassland landscape. The decrease in long-term grassland area slowed down in all agricultural regions from 2017 onwards.

In the Campine region, long-term grasslands were disappearing faster compared to other regions (Figure 9). Interestingly, the area of agricultural grassland decreased in all agricultural regions, but not in the Campine regions when comparing data from the LPIS geodatabase of 2018 with 2008 [20]. Thus, the observed decrease in long-term grassland occurred in non-agricultural grassland. In the Loam region, long-term grassland is decreasing rapidly compared to the other agricultural regions. The Campine region and the Loam region were also the two regions with the lowest area of long-term grassland.

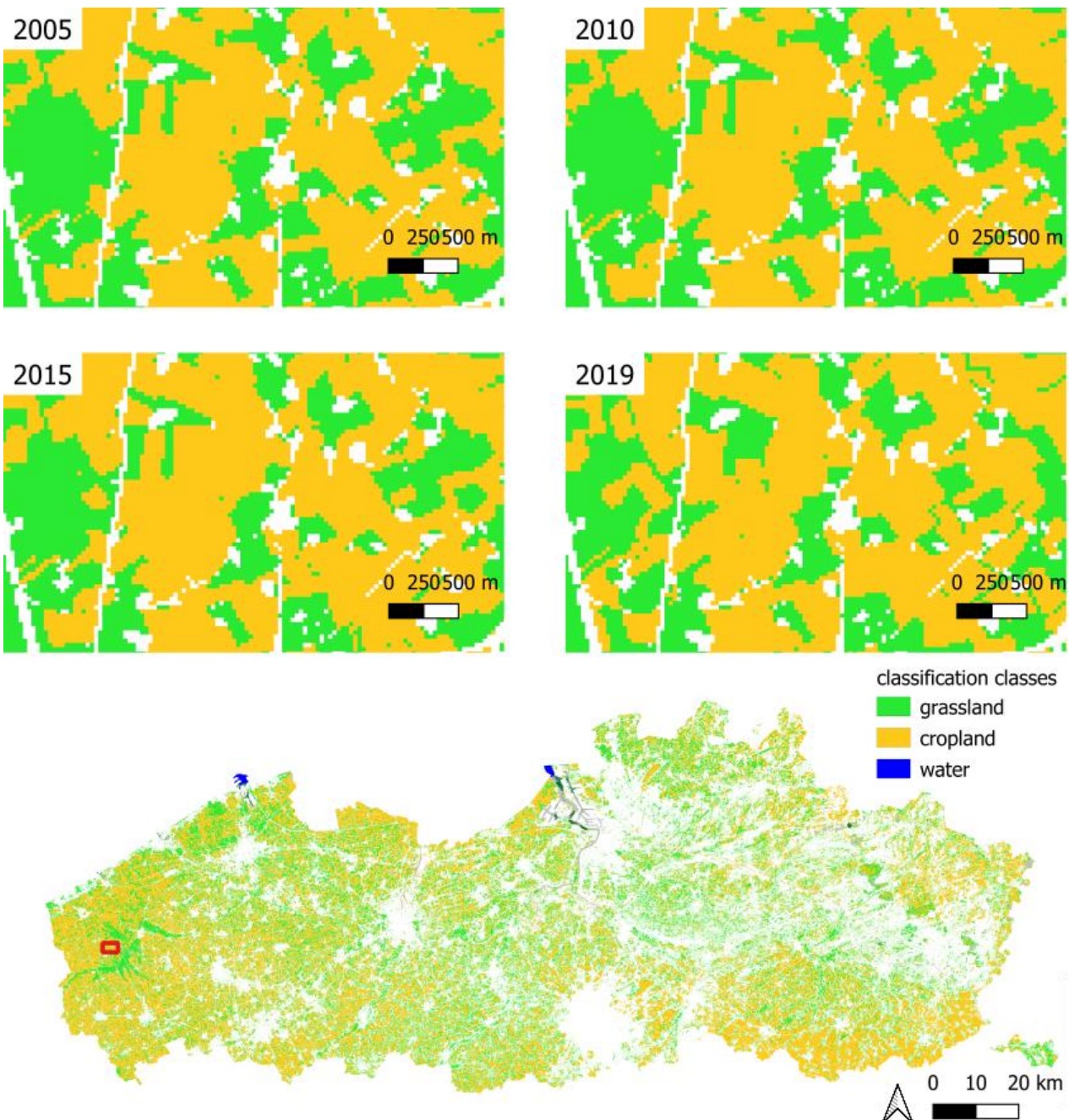

**Figure 2.** Snapshot of the historical land cover classification maps for the years 2005, 2010, 2015 and 2019 and overview of the classification product (2019). The red square on the map of Flanders indicates the snapshot area.

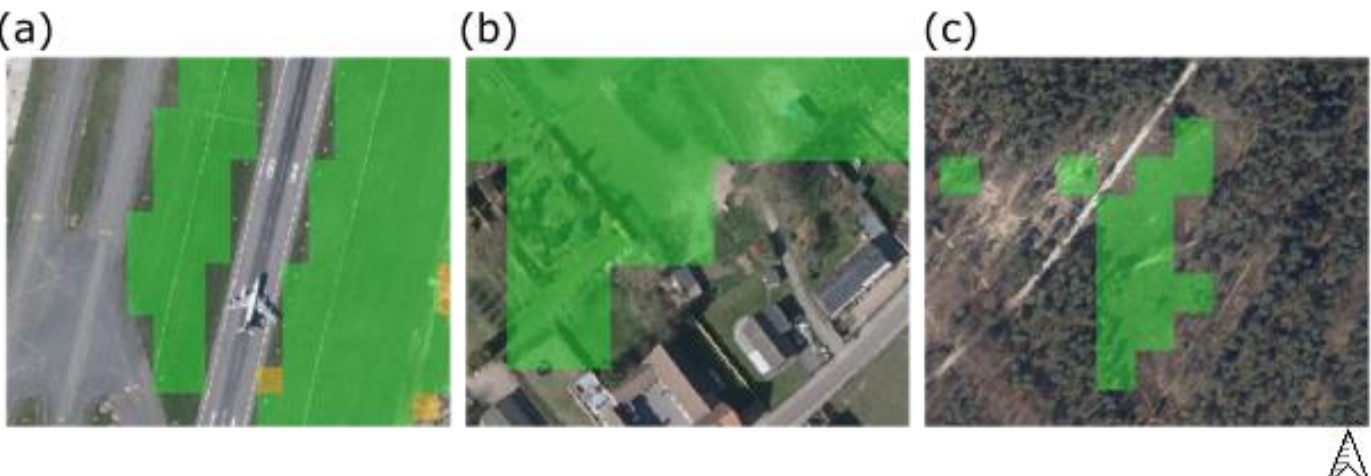

**Figure 3.** Examples of non-agricultural grassland correctly being classified by the historical land cover classification maps: (**a**) grass strips in airports, (**b**) gardens, (**c**) natural grasslands. Background orthophoto is from April 2021 [28].

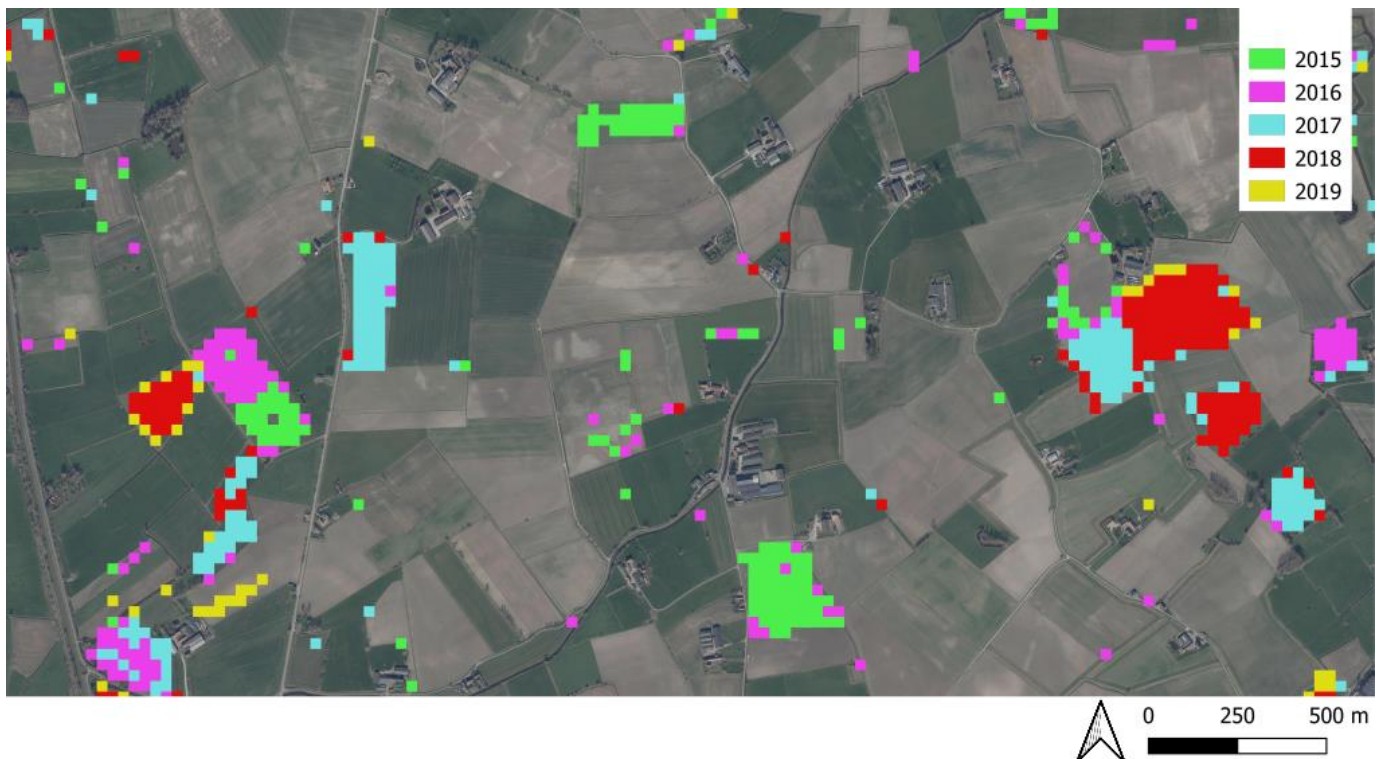

**Figure 4.** Location of grasslands that were grasslands from 2005 onwards but were converted to cropland between 2015 and 2019 according to the historical land cover classification maps. The colors indicate the year of conversion from grassland to cropland. Background orthophoto is from April 2021 [28].

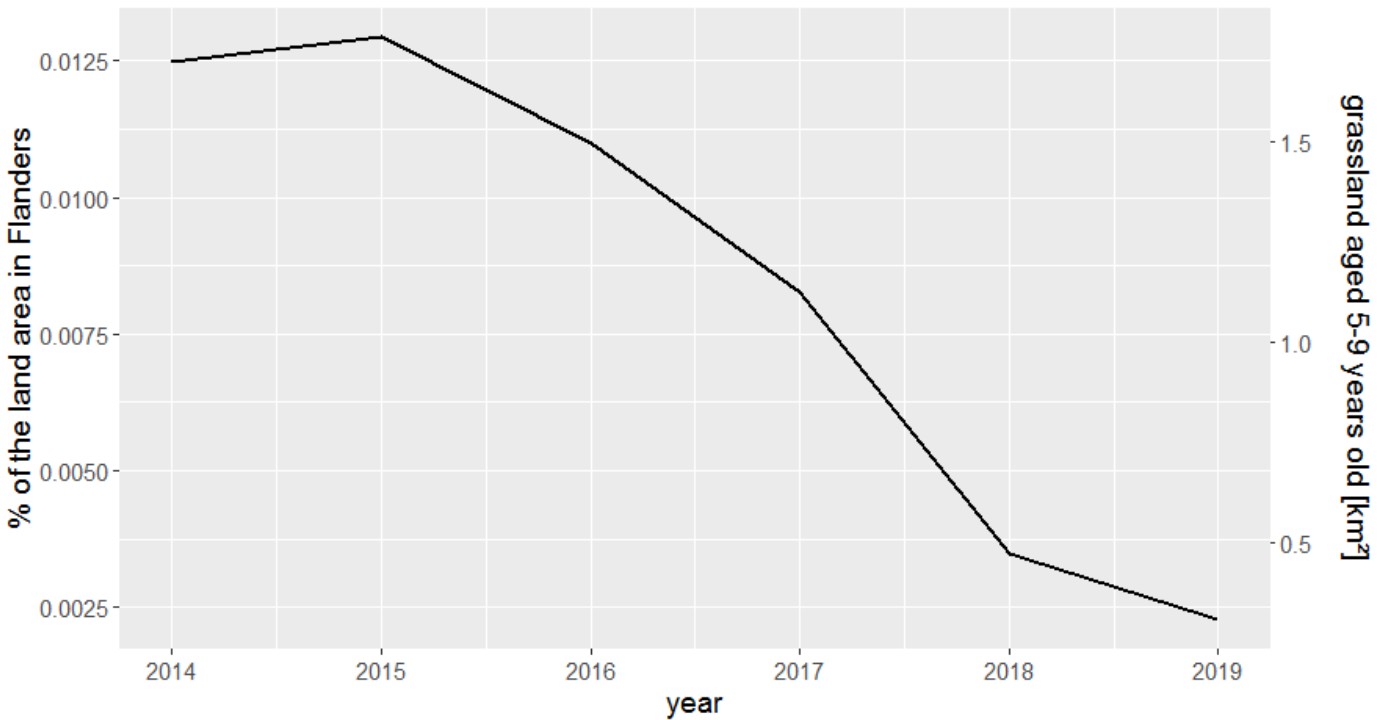

**Figure 5.** Area of long-term grassland aged 5–9 years old from 2014 to 2019 in Flanders, expressed as percentage of Flanders (left axis) and km$^2$ (right axis). The area of long-term grassland aged 5–9 years old was derived from the historical land cover classification maps from 2005 to 2019.

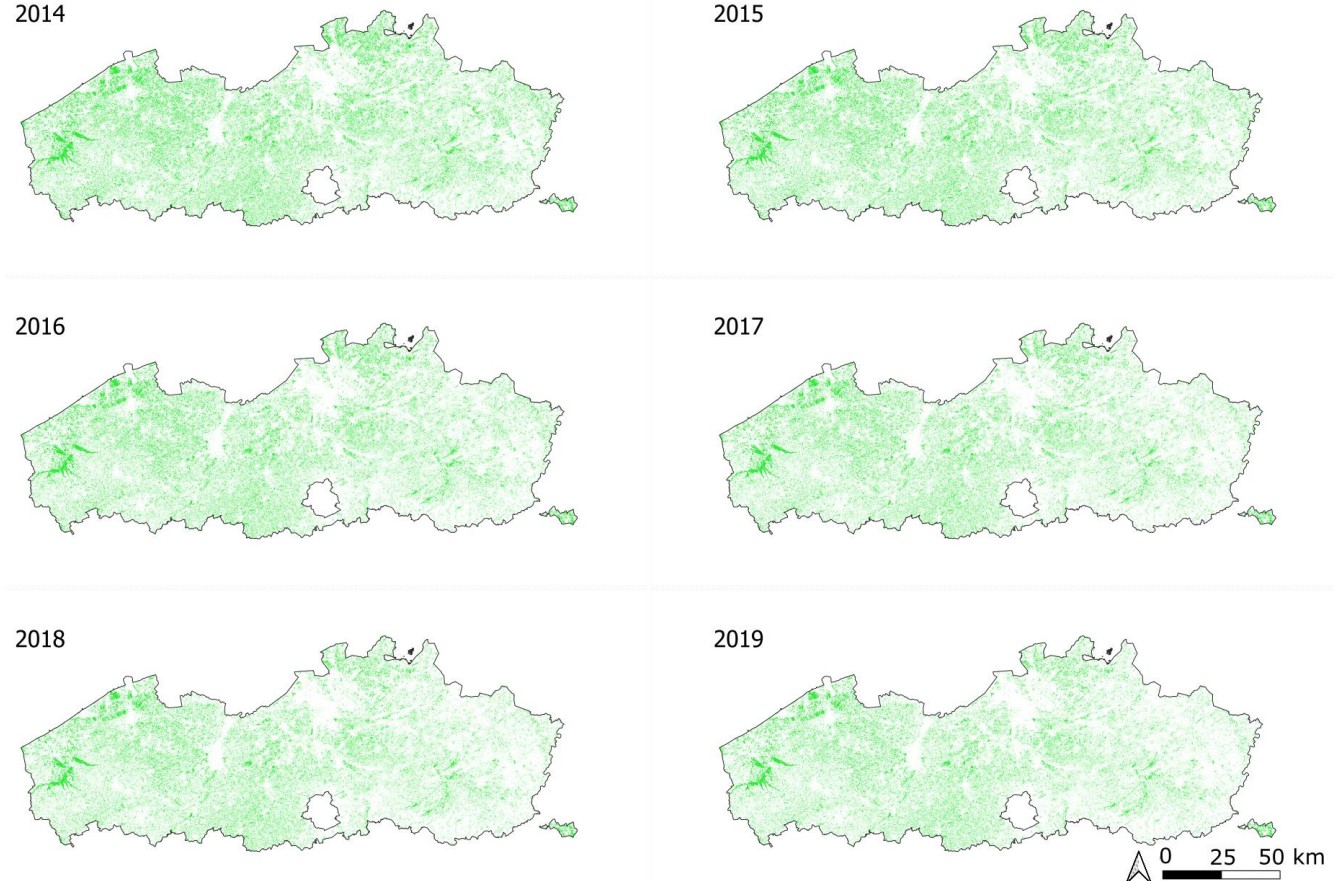

**Figure 6.** Map indicating the location of long-term grassland aged 10 years or more from 2014–2019.

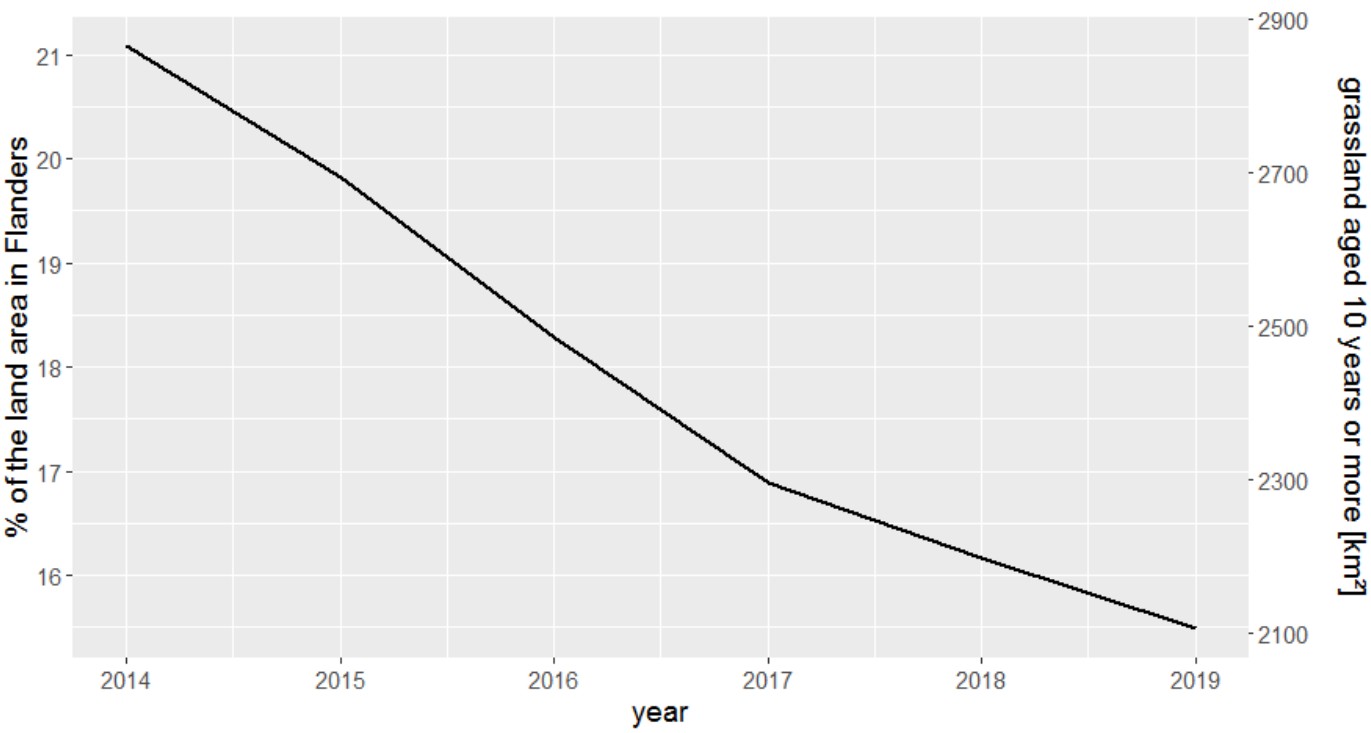

**Figure 7.** Area of long-term grassland aged 10 years or more from 2014 to 2019 in Flanders, expressed as percentage of Flanders (left axis) and km² (right axis). The area of long-term grassland aged 10 years or more was derived from the historical land cover classification maps from 2005 to 2019.

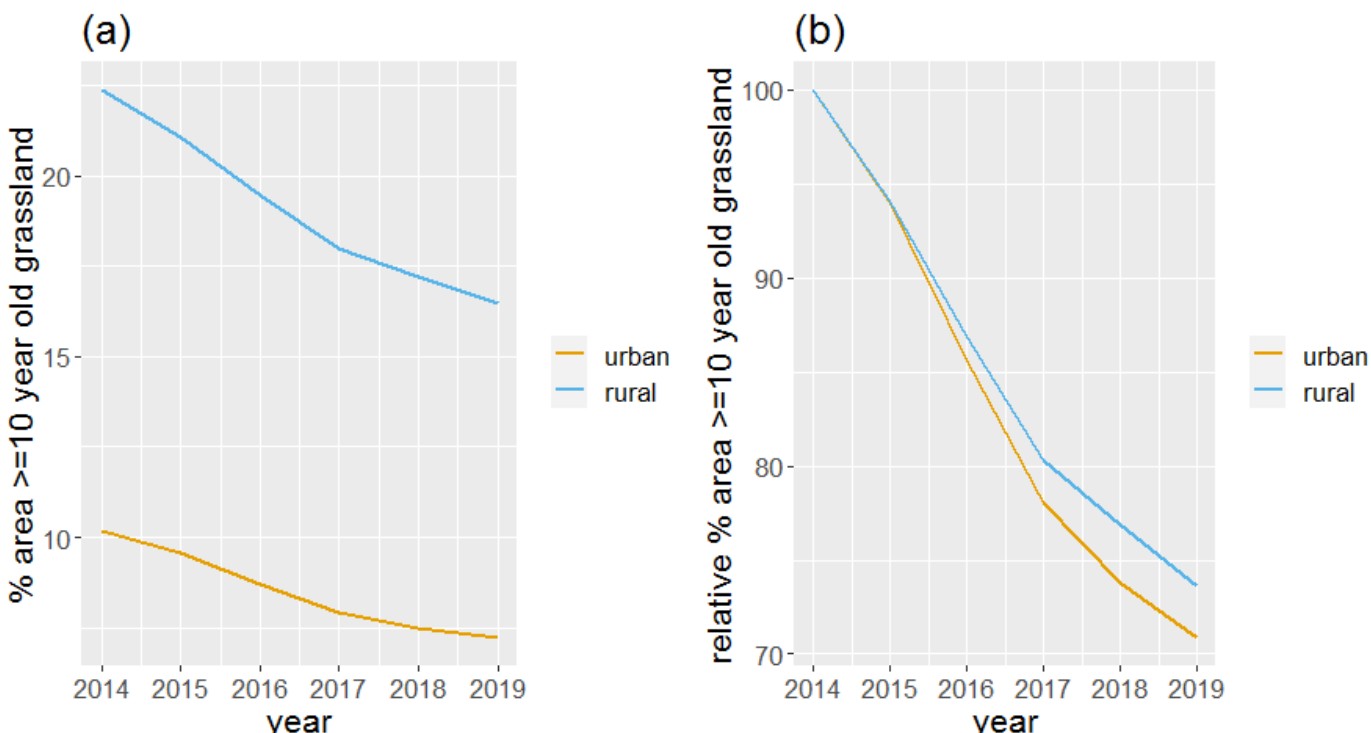

**Figure 8.** (**a**) Percentage of area long-term grassland aged 10 years or older from 2014 to 2019 in urban versus rural areas. (**b**) Relative % area change of long-term grassland from 2014 to 2019 in urban versus rural areas.

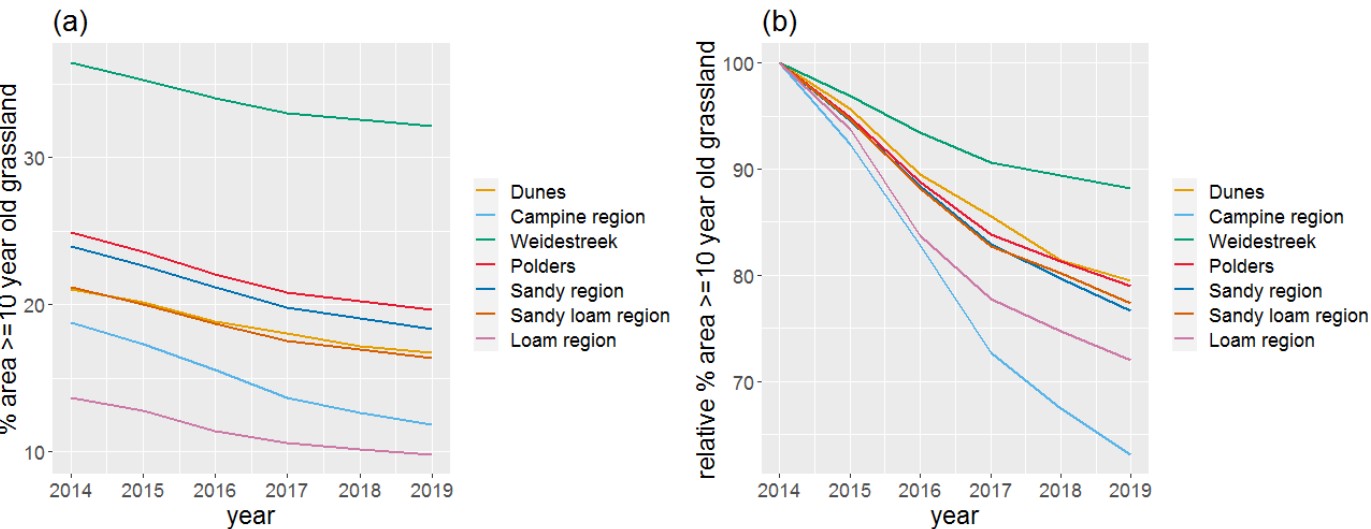

**Figure 9. Figure 9**. (**a**) Percentage of long-term grassland aged 10 years or older from 2014 to 2019 in the seven agricultural regions. (**b**) Relative % area change of long-term grassland from 2014 to 2019 in the seven agricultural regions.

## 4. Discussion

The historical land cover classification maps from 2005 to 2019 allowed us to evaluate the location of grasslands, the conversion of grasslands to arable land and the age of grasslands. A random forest classification algorithm was applied to obtain the historical land cover classification maps for Flanders, using an approach similar to [12,16]. When comparing the accuracies with the CGLOPS-100m database over Europe, which are the most consistent baseline land cover maps from 2015 up to present day at 100 m spatial resolution, consistently higher accuracies were achieved for 2015 in this study for the classes forest, arable land, and grassland. This is partly explained by the lower number of land cover classes that are considered in this study, but also by the increased amount of training data that was obtained from the LPIS database, which has a profound positive effect on the classification results. For 2012, significantly lower accuracies were obtained. Although still higher than the CGLOPS-100m validation results, the effect of the lower number of images obtained by Landsat 5 and 7 in the period 2004–2013 and the failure of the Scan Line Corrector significantly degrades the performance of the land cover classification algorithm. The application of the Hidden Markov Model to remove erroneous classification results and improve spatio-temporal consistency due to lack of accurate input information is therefore justified and has proven to significantly improve time series analyses in previous studies [12,29,30].

Information on the location of grassland converted to arable land (Figure 4) is useful for the implementation and evaluation of policies to protect long-term grassland. Applied to Flanders in Belgium, we found that 3% (449 km$^2$) of grasslands that had been grassland since 2005 were converted to arable land between 2015 and 2019 (Figure 4). This conversion of grassland to cropland affects the landscape and the ecosystem services it provides [6]. The European Common Agricultural Policy (CAP) stipulates that the ratio of agricultural grassland older than or equal to 5 years to the area of agricultural land must not decrease by more than 5%. Once this threshold is exceeded, there is a restriction on the conversion of grassland to arable land and farmers may be obliged to restore converted grassland. In addition, there is a specific regulation for farmers in Flanders that prohibits them from converting ecologically valuable grasslands to arable land, these grasslands are defined as long-term grassland of 10 years or more. The observed trend in the conversion of (agricultural and non-agricultural) grassland to arable land suggests that the current agricultural policies that aimed at preventing the conversion of grassland may not be sufficient in Flanders (Figure 4). Maps showing the conversion of grassland to arable land

(Figure 4) can be useful for identifying fields where grassland restoration is needed if the 5% threshold is exceeded.

The historical land cover classification maps prove particularly useful for studying changes in grassland older than 10 years, i.e., long-term grassland, in urban and rural regions and in sub-regions (Figures 5 and 7). The area of long-term grasslands in Flanders decreased by 6% between 2014 and 2019. Considering the ecosystem services provided and the area of long-term grasslands (15% in 2019 in Flanders), it is necessary to monitor this observed decrease in long-term grassland area and to develop appropriate policies. The importance of protecting long-term grassland from conversion to ensure the provision of ecosystem services has recently been highlighted [6]. The historical land cover maps can be a useful tool to gain insight into the current situation of long-term grasslands and to identify the regions or sub-regions where urgent action is needed.

Urban areas have a high demand for green spaces such as long-term grasslands and could benefit from their ecosystem services such as biodiversity [31]. We found that long-term grasslands were less common in urban areas than in rural areas in Flanders (Figure 8). Furthermore, the area of long-term grasslands decreased faster in urban areas than in rural areas between 2014 and 2019 (Figure 8). The area of long-term grasslands also decreased in all subregions, but with different dynamics (Figure 9). The location of long-term grasslands, whether agricultural or non-agricultural, and their rate of disappearance can be monitored using the developed methodology and enable the detection of hotspots of change. Appropriate policies to protect and monitor long-term grasslands at the local level should be high on the political agenda.

## 5. Conclusions

We used a random forest classifier to produce a series of Landsat-based land cover maps from 2005 to 2019 for the region of Flanders, Belgium, and concluded these historical maps were useful for evaluating grassland area and age. This information can be used to assess compliance with existing regulations or highlight the need for regulatory adjustments. Using Landsat-based historical classification maps, we were able to locate grasslands that were grassland between 2005 and 2014 but were converted to arable land between 2015 and 2019 and identify the year in which these grasslands were converted. Since 2003, the European cross-compliance regulation has required farmers receiving CAP income support to maintain grasslands at least 5 years, referred to as permanent grasslands in the CAP [11]. Compliance with this permanent grassland protection regulation is evaluated at regional or national level. However, land use changes can still occur at farm and sub-regional level in EU Member States [11]. Our analysis for Flanders, Belgium, has clearly shown a general decrease in the area of long-term grasslands, yet significant regional differences in the conservation of these important biotopes have been identified. Our study demonstrates that multi-annual high resolution satellite imagery provides objective and quantitative information on long-term grasslands to support climate, agricultural, environmental, and nature policies.

**Author Contributions:** Conceptualization and methodology, A.G., N.S., J.D. and A.V.; validation, formal analysis, writing—original draft preparation, A.V., N.S. and J.D.; data curation and writing—review and editing, A.G., N.S., J.D. and A.V.; supervision, A.G., funding acquisition, A.G. All authors have read and agreed to the published version of the manuscript.

**Funding:** The authors acknowledge funding from the European Union's Horizon 2020 Research and Innovation Programme under grant agreement No. 818496 and No 818187.

**Data Availability Statement:** Landsat 5, 7, and 8 area available on Google Earth Engine.

**Conflicts of Interest:** The authors declare no conflict of interest.

## Appendix A. Location of the Scenes (Path/Row)

| Path | Row |
|------|-----|
| 200 | 25 |
| 200 | 24 |
| 199 | 25 |
| 199 | 24 |
| 198 | 25 |
| 198 | 24 |
| 197 | 25 |
| 197 | 24 |

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
