# Peer review of "Spatio-Temporal Dynamics in Grasslands Using the Landsat Archive"

_land, doi:10.3390/land12040934_

Round 1
Reviewer 1 Report
“Spatio-temporal dynamics in grasslands using the Landsat archive,” I think the manuscript is interesting and provide important baseline information on changes in grassland coverage in Belgium. Authors have quantitatively described changes in grassland cover using Landsat images. However, the following concerns were noted and that should be taken into account before acceptance of the manuscript.
The Abstract appears inflated, it is not well structured, and too much unnecessary information thus I recommend rewriting the abstract.
The content of the introduction is appropriate however, the arguments are not well structured. Additionally, it is not unclear what the authors are meant by changes in grassland. For example, “(ii) evaluate the change in the area of long-term grassland aged 5-9 years and 10 years or older”. Changes could happen qualitatively and quantitatively so such information should be clearly described.
The description of the methodology is appropriate; however, please state image acquisition dates in a table. Important to have acquisition days of images, path/row of images, and cloud cover of data. The reader would like to know whether authors have used the same growing season when they acquire images.
In the results section:
Line 138-139 The land cover classification maps produced from 2005 to 2019 were used to evaluate the evolution in grassland extent and conversion of grassland to arable land. It is not clear what is meant by evolution. Please elaborate on it.
The North direction should be indicated on each map.
Graph axis:km2 grassland aged 5-9 years old; it should be grassland aged 5-9 years old (km2 ) hence all axis should be correct accordingly.
Discussion and conclusions are appropriate, however, concluded information is rather general, especially conclusions
Line 332: Flanders in Belgium clearly shows a decline in long-term grasslands; what is meant by decline? Area or other quality.
Overall, the language of this manuscript should be extensively improved. Many issues were noted in terms and wording. The organization and some of the writing need some clarification and rewording. I recommend that the manuscript be re-reviewed before publication
Reviewer 2 Report
Reviewer's Comments
Title: Spatio-temporal dynamics in grasslands using the Landsat archive
Manuscript ID: land-2302973
General Remarks
Generally, the article land-2302973 entitled " Spatio-temporal dynamics in grasslands using the Landsat archive " demonstrates one of the widely used technique (analysis of remotely sensed data) to study change detection of grasslands dynamics. The manuscript is clearly written, with some Minor comments.
1- There are numerous language issues (sentence structure, grammatical errors, and typos) in this manuscript. The entire document has to have its language edited carefully.
2- In the title of manuscript insert the country of studied area.
3- The figures need to be more quality in all manuscript
Reviewer 3 Report
Title: Spatio-temporal dynamics in grasslands using the Landsat archive
This is an interesting paper, sot the mapping of extensive regions.
Abstract:
The approach is straight forward, regarding the objectives. However, no information about 1) the methodological approach or 2) the nature of the datasets (not only the name of the missions, but the sensors or their characteristics) is provided.
Editing is required, for some of the text and the repetition of the therm “grasslands” should be moderated in some sentences ej:
“However, there is currently no multi-annual information on the location of grasslands at a high spatial and temporal resolution which hampers the monitoring of changes in grassland area and their characteristics such as age to evaluate and adjust the grassland protection regulations”
Define “long term grassland”.
Introduction:
Just as in the abstract, define long term grassland, what are them? And why the temporal threshold of 5 years or older?
L56-57: this sentence “In the EU-6 around 7.1 million ha (or 30% of the 1967 level) of long-term grasslands disappeared between 1967 and 2007” sound important, but it is incomplete since it does not provide a follow up. Complete the idea…
L78-79: this sentence is confusing “Using the Landsat archive [13] was able to evaluate 78
changes in the Brazilian grassland area from 1985 to 2017” who? For what? Lack of context.
L88: this sentences “Currently it is unclear what the age is of the agricultural grasslands that have disappeared” is this part of the knowledge gap addressed? Then it should be the start of another paragraph, since it is key.
Some edition is needed for the section, but the objective is clear.
Methods:
There is no context on the region of study. I think this section should be added, so we understand the heterogeneity of the area, and what can be found in there. Why is it important? what is the extent? Etc…
What was the level of processing of the imagery used? What product?
Given the SLC issues with ETM+, why did the authors chose to use it instead of TM (Landsat 5)?
Accuracy protocol should be clearly stated.
Time step of the series should be stated (does not matter if it is year by year).
Change approach should be clearly stated (it look like a post classification change detection, but is not obvious).
Did the authors used all images for each year for the classifications? Phenology?
Location of the scenes (Path/row) used?
Figure 1: should be clearer, what are the transparent polygons?
Results:
Table 1: confusion matrix, are typically presented as number of readings per cover type. I suggest a change, since percentages does not tell the complete history.
Overall accuracies?
Round 2
Reviewer 1 Report
Figure-05: The axis should correct as % of the land area in Flanders. Otherwise, it is a bit awkward
The figures are not much clear so need to improve the clarity of the graphs.
Author Response
Figure-05: The axis should correct as % of the land area in Flanders. Otherwise, it is a bit awkward
- Axes were changed in figures 5&7
The figures are not much clear so need to improve the clarity of the graphs.
- font and line size of figures were improved
Reviewer 3 Report
Figure 1: Add the ag fields to the label instead then clarify on the footnote.
Author Response
Figure 1: Add the ag fields to the label instead then clarify on the footnote.
- Figure 1 was adapted in legend label